# Numerical Study on Seismic Behavior of Flexural Frames with Semi-Rigid Welded Steel Connections Considering Static and Reciprocating Loads: A Performance-Based Earthquake Approach

**Majid Malekzadeh** [1] **and Mohsenali Shayanfar** [2,*]

1   Structural Engineering, Iran University of Science and Technology, Tehran 16846, Iran; malekzadeh@guilan.ac.ir
2   Faculty Member of Civil Engineering Group, Iran University of Science and Technology, Tehran 16846, Iran
*   Correspondence: shayanfar@iust.ac.ir

**Abstract:** This paper aims to apply a performance-based earthquake engineering approach to assess the assurance of flexural frames whose members are jointed together by using new modified RBS connections, namely, semi-rigid welded steel connections, which obey a progressive failure mechanism. First, the structural members and connections are modeled and predesigned in ETABS software, and then, using OpenSees software, a series of nonlinear progressive failure analyses are performed on the built models extracted from ETABS. To this end, three types of multi-story structures with 3, 10, and 15 are modeled. The models are subjected to 15 earthquakes, such as Northridge (1994), Kobe (1995), Chichi (1999), Bam (2003), Tabas (1978), and so on. The connections are modeled by a series of rotational springs whose nonlinear behavior is estimated by a three-line curve that is established based on the modified Ibarra–Krawinkler deterioration model. Finally, obtaining the maximum ground acceleration versus the maximum relative drift of the floors, the fragility curves of the structures for a collapse level (CP) are determined, through which the seismic performances of the models are evaluated. The results show that by reducing the number of structural floors, the ductility of structures was reduced, and by increasing the ductility of structures, higher drifts in structures were achieved at the same seismic level. The average amount of ductility reduction coefficient in structures with RBS was 1.06 times those without RBS, which indicates that the energy dissipation capacity in structures without RBS connection is higher than in those with RBS. Local analysis of connections shows a 9% increase in the plastic rotation capacity if RBS connections are used. The ductility of all frames with RBS connection increased slightly compared to frames without RBS.

**Keywords:** reduced beam section (RBS); seismic behavior; incremental dynamic analysis (IDA); performance-based design

## 1. Introduction

The bone joint is one of the modern welded connections (Reduced Beam Section, RBS), which has been considered by structural designers in countries such as the United States and Japan in seismic zones, after the Northridge (1994) and Kobe (1995), in order to find a solution for increasing the ductility and limiting the failure modes. Steel bending frames are designed to be able to lose a lot of energy due to flow and plastic deformations. To absorb significant energy from an earthquake, the occurrence of yield and plastic hinge formation is essential, and the brittle failure of the connection prevents the welded flexural connections from exhibiting the inelastic behavior desired to withstand earthquake loads. During the three past decades, many studies [1–24] attempted to examine the behavior of various connections, including the RBS connection in steel structures, by performing both numerical and experimental analyses, and the main scopes and findings from these studies can be concluded as follows.

Barakat and Chen (1991) evaluated the behavior of semi-rigid frames without bracing by performing a simple analysis method [1]. By examining improved methods for constructing a realistic coded and finite structure, Hsieh and Deierlein (1991) proposed a method for combining nonlinear bonding behavior in the analysis of three-dimensional structures. They implemented a connection model by which the nonlinear moment response for rotation in the direction of large and small axes is evaluated [2]. Barakat and Chen (1990) attempted to develop a simple method to assess the nonlinear response of three-dimensional steel frames with semi-rigid connections based on previously developed AISC/LRFD design methods [3]. To evaluate the performance of semi-rigid connections, Shayanfar et al. (2017) improved the first-order structural basis estimation by Monte Carlo simulations [4]. Thomas and Kurt (1992) investigated the reliability of the welding behavior of semi-rigid frame connections. By performing a series of tests on the connection models and obtaining moment–rotation curves from these tests, they constituted a database for determining the probability states of reliability that can be expected with the specific behavior of these connections [5]. Performing model tests, Wai and Kishi (1989) established a database of semi-rigid and rigid steel column beam connections at Purdue University, by which they developed a Data Base (SCDB) program to analyze the steel connections [6].

Moreover, Kishi and Wai (1990) studied the moment–rotation relationships of angular semi-rigid connections [7]. Farzaneh et al. (2012) demonstrated the extent of damage to the connections of steel buildings, after the 1994 Northridge earthquake, as well as major deficiencies in the design and fabrication of special steel bending frames, which is directly related to the connection performance. They showed that the connection of the beam clamp to the box column is an effective connection in transferring the plastic beam hinge away from the column and due to satisfactory ductility [8]. Saniee et al. (2011) evaluated the seismic behavior of I-shaped beam connections to can columns in special bending frames [9]. Lyse and Gibson (2001) investigated the effect of welding angles on column beam connections [10]. Okura and Fukumoto (1993) investigated analytically and experimentally local stresses in cross-beam flange connections. They point out that local stress should be considered in the design of fatigue [11]. Deylami and Tolo Kian (2014) studied the cyclic bending behavior of flexural connection of an I-shaped beam to a box-shaped column using welded and flexible wing sheets. They showed that, due to the flexibility of the connection, the stress in the groove connecting the joining sheets of the beam to the wing of the column reduces significantly; however, the overall behavior of the connection is not affected significantly [12]. Colson (1991) presented a one-dimensional theoretical modeling to study semi-rigid connection behavior [13]. Elnashai and Elghazouli (1994) studied the behavior of semi-rigid and rigid connections under a seismic loading condition [14]. Jihong and Xu (2017) used the member discrete element method to study the seismic behavior of steel frames with semi-rigid connections. Performing fracture analysis, they showed that semi-rigid and solid steel frames have more capacity compared to hard and rigid steel frames [15].

Furthermore, Silva et al. (2015) performed nonlinear transient analysis on the planar steel frames with semi-rigid connections. They showed that the hysteresis curve of the connections has an important effect on the response of frames and is an important source for inhibition during structural vibration [16]. Shen J. et al. (2015) evaluated the seismic behavior of two steel buildings with rigid and semi-rigid composite frames that resulted in the construction of semi-rigid internal frames that can lead to less shear and fewer columns and connections and increase the lateral bearing capacity of the building [17]. Krolo et al. (2015) compared the nonlinear seismic response of steel frames with semi-rigid and complete-rigid connections. [18]. Beheshti et al. (2019) implemented the cloud analysis approach to evaluate the seismic reliability of a bending steel frame with two different ductility levels [19]. Beheshti et al. (2017) investigated the performance of two conventional methods for seismic improvement of flexural strength concrete and steel structures. They implemented the force control method to achieve fragility curves using OpenSees software [20]. Bahramirad et al. (2015) investigated the formulation of incremental nonlinear dynamic analysis using an

expanded overlay analysis mechanism under near-fault excitation [21]. Fayun et al. (2017) evaluated the accuracy of seismic analytical response using nonlinear numerical simulation by using the FAMA recommendation approach [22]. Marijana and Tanja (2017) explored seismic damage perspectives using incremental nonlinear analysis and fragility curves in high-rise structures [23]. Chenfeng et al. (2019) studied the ultimate strength of welded stiffened plates by using numerical modeling [24].

This study aims to investigate the linear and nonlinear behavior of semiconductor welded steel connections with a combination of static and reciprocating loads. The effect of semiconductor welded steel connections on the seismic behavior of three multi-story steel frames is investigated. The main innovations of the present study can be concluded as follows. First, as mentioned above, since steel bending frames with semi-rigid connections are one of the systems resistant to lateral loads such as earthquakes and wind, plastic connections are formed in several parts of the structure. Therefore, the role of connection strength in these frames remains a crucial and complex problem needing more in-depth investigation. Besides, in this study, a new type of RBS connection, namely, semi-rigid welded steel connection, is used for modeling, on which very few studies have been conducted before this. Furthermore, in this study, the analytical equation of semi-rigid connection (modified RBS connection) is expanded to estimate the energy absorption of this type of connection in relation to the moment–rotation relationship.

## 2. Methodology

This research is presented in two main parts (i.e., Sections 3 and 4). Section 3 presents the linear and nonlinear numerical analysis on some flexural frames whose members (i.e., beams and columns) are joined by implementing a new type of semi-rigid connection (modified RBS connection). To this end, first, three types of three-dimensional steel flexural structures with simple connections and with 3, 10, and 15 floors (Figures 1–4) are modeled in ETABS 2016 software in which preliminary dimensions of the frame members (i.e., beams and columns) are determined by performing pre-design analysis. Then, some frames of 3D models in built in ETABS are selected and transferred to the OpenSees ver. 2.5 software in which the simple connections are replaced with the semi-rigid welded steel connections. In this step, the seismic responses of semi-rigid welded steel connections are investigated. To this end, the selected frame systems are subjected to both the static and reciprocating loads, and using the effects of 15 different earthquakes such as Northridge (1994), Kobe (1995), Chichi (1999), Bam (2003), Tabas (1978), and so on, the seismic responses of connection models are investigated. The connections are modeled as a series of nonlinear moment-rotational springs. A three-line curve is adopted to represent the constitutive behavior of the springs. The characteristics of this constitutive curve (in other words, the slopes of a three-line curve that represents the nonlinear stiffness of the modeled springs) are extracted by using modified Ibarra–Krawinkler deterioration relationships. At end of this section, the obtained results for the seismic response of the studied connection are discussed and a comparison is made with a previously published work. Finally, in part 2 (Section 4), the effects of increasing failure in steel towers are evaluated by using a performance-based design method. To this end, first, the variation of the maximum relative drift of the floors versus maximum ground acceleration is obtained, based on which the fragility curves of the structures for a collapse level (CP) are determined.

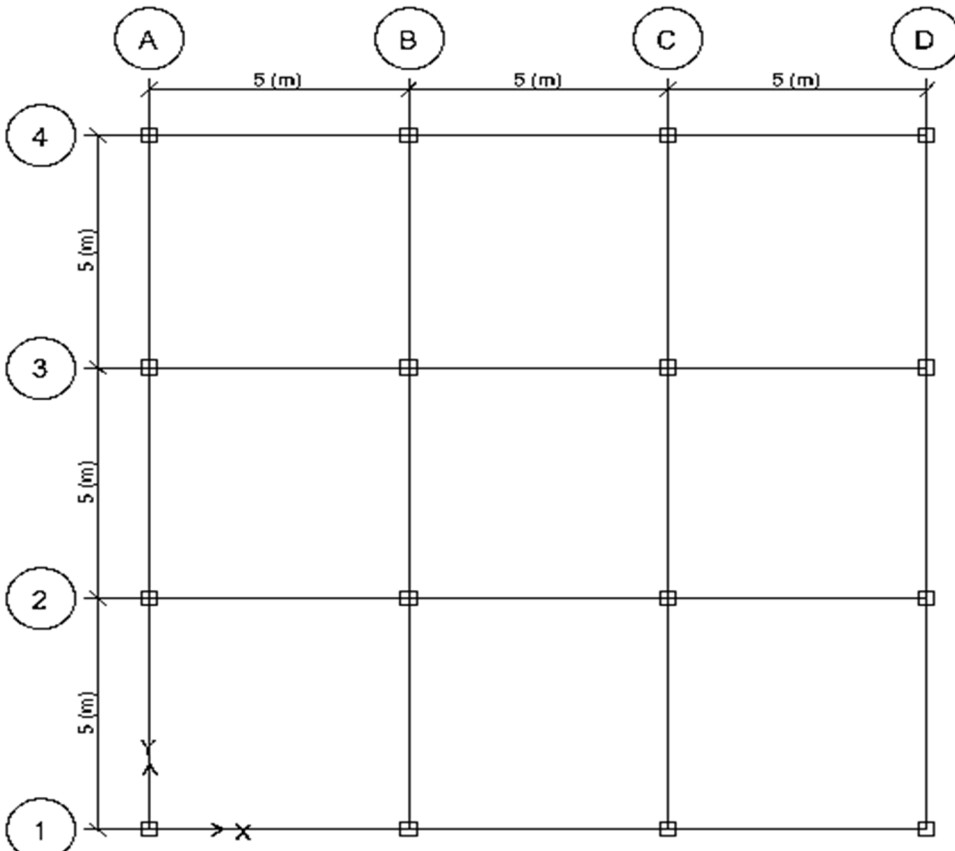

**Figure 1.** Modeled buildings' plan in the Etabs software.

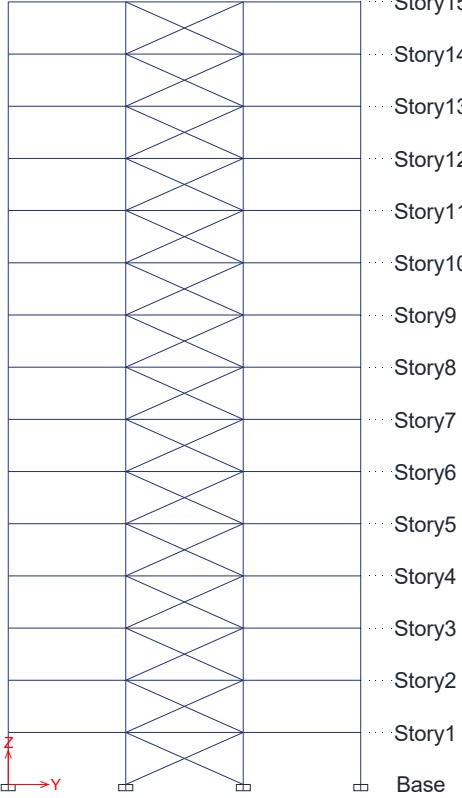

**Figure 2.** 15 Story building frame in the Etabs software.

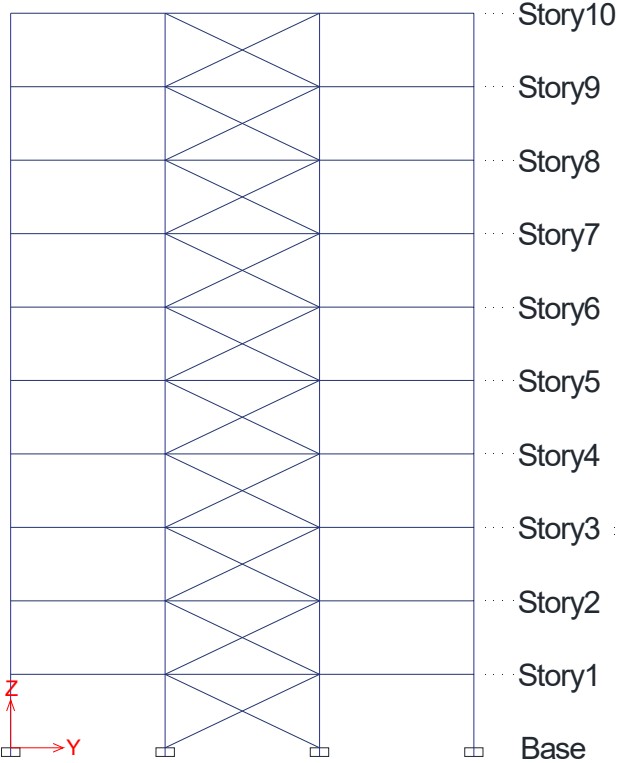

**Figure 3.** 10 Story building frame in the Etabs software.

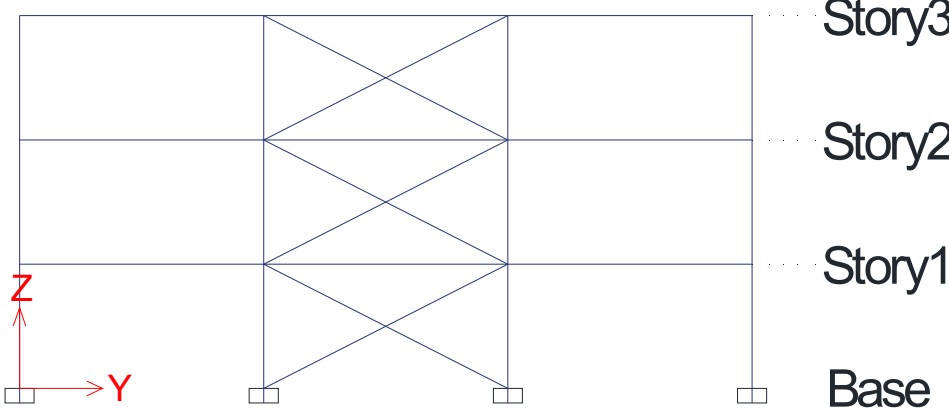

**Figure 4.** 3 Story building frame in the Etabs software.

## 3. Seismic Numerical Analysis

### 3.1. Modeling in ETABS

As explained in the methodology of research, first, three types of three-dimensional structures models are built in ETABS. The specifications of the structures modeled in ETABS software are presented in Table 1. Figures 1–4 also show the three types of structure models built in the software. The preliminary dimensions of the beams and columns determined through pre-design analysis in the software are listed in Table 2.

**Table 1.** Details of the structures considered in this research.

| No. | Number of Story | Position of Buckling Braces | Bays' Width (m) | Height (m) | Type of Resistant System |
|---|---|---|---|---|---|
| 1 | 3 | All floors have a bracing system and simple connections | | =3 × 3.2 = 9.6 | Medium steel moment frame with special convergent bracing |
| 2 | 10 | | | =12 × 3.2 = 32 | |
| 3 | 15 | | 5 | =12 × 3.2 = 48 | |
| 4 | 3 | All floors have a semi-rigid bracing system (RBS) | | =3 × 3.2 = 9.6 | |
| 5 | 10 | | | =12 × 3.2 = 32 | |
| 6 | 15 | | | =12 × 3.2 = 48 | |

**Table 2.** Used sections in each selected buildings' frame (for two types of connections in this research).

| 15 Story building, with bracing system in all stories | | | |
|---|---|---|---|
| Story Level | Column | Beam | Brace |
| 1–5 | Box 700 × 700 × 40 | IPE 400 | Tubo 160 × 160 × 10 |
| 6–10 | Box 600 × 600 × 35 | IPE 360 | Tubo 140 × 140 × 12.5 |
| 11–15 | Box 450 × 450 × 25 | IPE 330 | Tubo 120 × 120 × 12.5 |
| 10 Story building, with bracing system in all stories | | | |
| Story Level | Column | Beam | Brace |
| 1–5 | Box 600 × 600 × 35 | IPE 330 | Tubo 160 × 160 × 10 |
| 6–10 | Box 500 × 500 × 30 | IPE 300 | Tubo 140 × 140 × 12.5 |
| 3 Story building, with bracing system in all stories | | | |
| Story Level | Column | Beam | Brace |
| 1–4 | Box 220 × 220 × 15 | IPE 270 | Tubo 120 × 120 × 10 |

*3.2. Modeling Procedure in OpenSees*

3.2.1. Types of Connections Used for Modeling

In this study three types of connections, shown in Figures 5–8, which are commonly used in practice for achieving the desired ductility, were used for the connections of selected frames. Figures 5–8 show the models of implemented connections in Open Sees software. The Schematic view and the OpenSees model of first-type connection are shown in Figures 5 and 7, respectively. This type of connection, which is the center-to-center nonlinear model connection, has been used in modeling the dual bracing structure. This type of modeling is modeled without considering the loss of stiffness and strength, and considering the connection source. These models allow for submission in beams and columns. For this purpose, concentrated plasticity is usually used as a torsion spring at the end of the beam or column. In this model, the spring remains rigid until the member surrenders to the moment, but then the strain stiffening behavior, which is defined as a percentage of the initial stiffness ($\alpha_s$), controls the behavior of the member. It is shown that $\alpha_s = 0.03$ gives acceptable results in calculating the relative displacement of the floor. Then, after reaching the maximum moment, the assigned behavior curve can be in a straight line (zero slope). Considering this, for designing new structures or evaluating existing structures, two criteria for accepting member strength and structural stiffness (relative displacement of the floor) should be controlled, but linear modeling from center to center is acceptable for designing dual bracing frames.

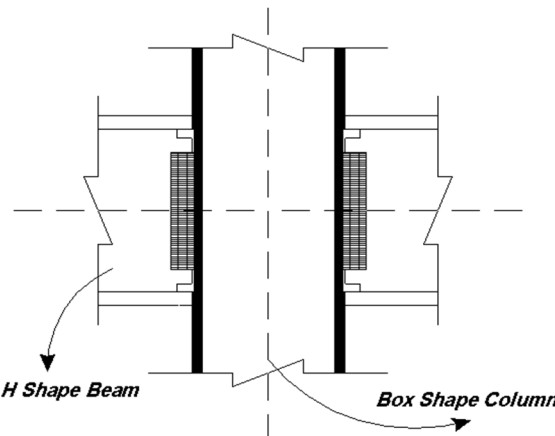

**Figure 5.** An overview of an example of a simple connection used (conventional connections before the Northridge earthquake).

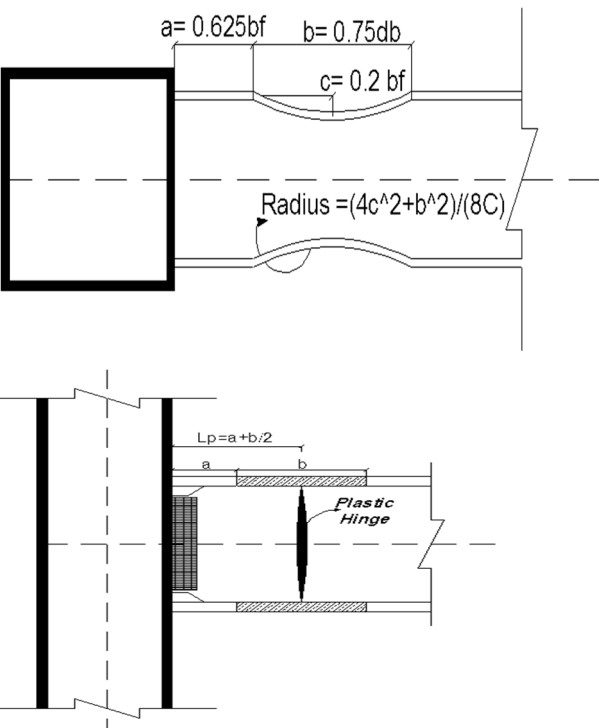

**Figure 6.** An overview of an example of the RBS-type connections used (conventional connections after the Northridge earthquake).

The Schematic view and the OpenSees model of second-type connection are shown in Figures 6 and 8, respectively. This type of connection, which is used in the frame with low and high ductility, is the non-linear model type with the connection spring in mind. In this model, the dimensions of the Weston beam are considered as the dimensions of the connection spring to assign a three-line behavior. A three-line torsion spring or two two-line torsion springs can be used in one of the connection corners. In order to achieve low ductility and less energy absorption (corresponding to the connections of structures before the Northridge earthquake), a Welded Unreinforced Flange with Welded Web connection (WUF-W) is assumed. In this type of connection, the beam wing is connected directly to the column wing using full penetration welding. Moreover, the beam die is connected to the beam die by corner welding with a corner weld and to the column wing by penetration welding. The rotational capacity specified in the regulations for the initial reduction in

the initial resistance is equal to the following equation. This type of connection is used in structures with a special bending frame system.

$$\theta_{sd} = 0.051 \tag{1}$$

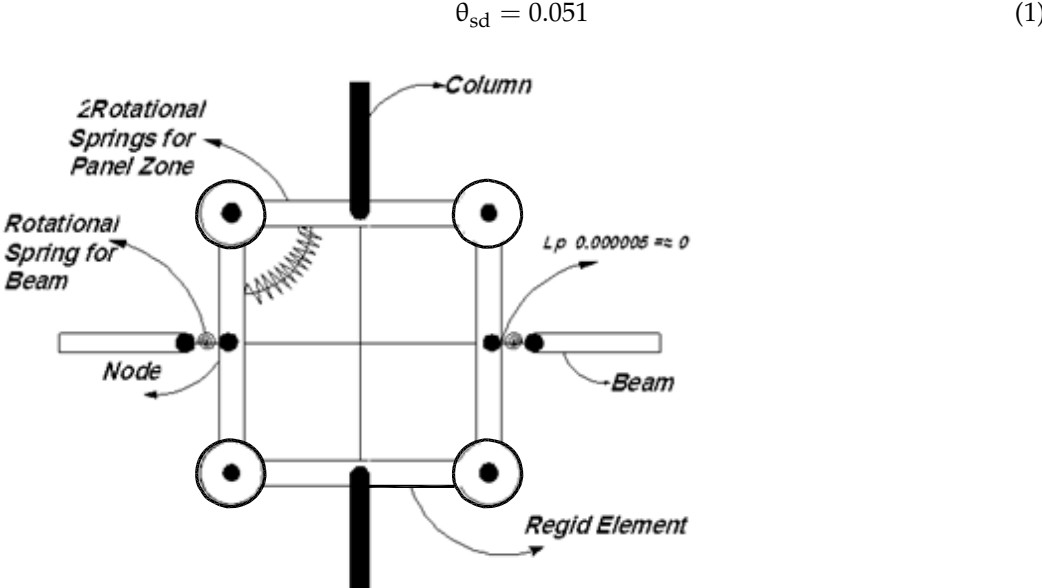

**Figure 7.** Schematic view of the Panel-Zone element used in modeling simple connections before the Northridge earthquake.

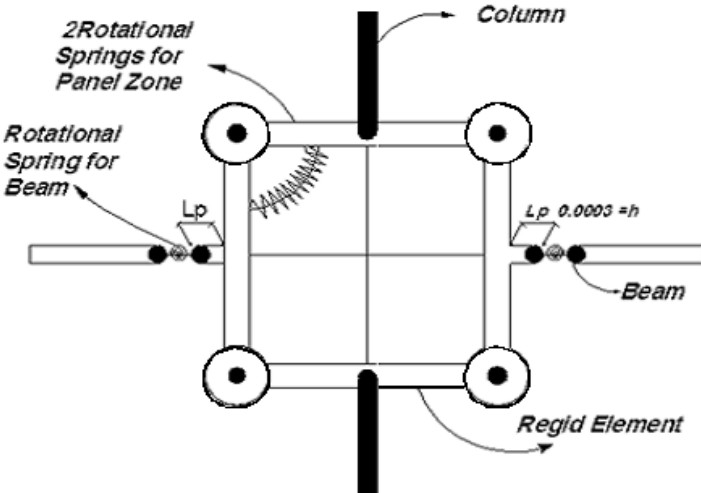

**Figure 8.** Schematic view of the Panel-Zone element used in modeling connections after the Northridge earthquake (corresponding to the RBS connection).

On the other hand, in order to achieve high ductility and absorb more energy (corresponding to the connections of structures after the Northridge earthquake), other types of connections, such as RBS have been used, which act as semi-rigid connections. This type of connection is also used for structures with a special bending frame system. This type of connection was also used after the 1994 Northridge earthquake, in which the connection is stronger than the beam. This type of connection is created by separating a part of the upper and lower wings of the beam at a certain distance from the connection point of the beam to the column. By doing this, the reduced area of the cross-section of the beam is the area that surrenders due to force. The reduced area of the beam wing acts as a factor to prevent sudden connection failure. The rotational capacity is defined according to the FEMA350

guideline for starting the initial resistance reduction mode based on the following equation where the height of the beam $(d_b)$ is in inches [25].

$$\theta_{sd} = 0.06 \sim 0.0003 d_b \tag{2}$$

On the other hand, due to the high lateral stiffness of these braces, the drift of the floors is easily responsive to service loads, but in comparison with the flexural frame system, it has a slight addition of resistance in the lateral capacity. In all the models studied in this paper, according to the recommendation of Spacon (1996), the inelastic beam-column element in OpenSees software has been used [22]. In the mentioned paper, force-based frame elements with distributed plastics and cross-section discretization in the form of fiber modeling have been used. In this model, in order to consider the possible buckling under axial loads, an initial defect in the system geometry is used in the form of an initial curvature. Here, the initial defect value of 0.08% of the initial limb length is placed at the end of the limb [22]. The Panel-Zone element has been used to model the connections in the bent frames. As mentioned before, the fiber model has been used in modeling the connections for the bracing frame, and the modeling of the connection bracing frame has been modeled in a simple way. It should be mentioned that, as mentioned before, in order to model the joints in OpenSees software, the length of the plastic joint has been used as a criterion for applying rigidity and semi-rigidity. Thus, in modeling the semi-rigid RBS connection, the length of the plastic joint equal to 0.0003 h has been used, and in modeling the WUF-W connection, the length of the plastic joint equal to 0.000005 has been used.

### 3.2.2. Modified Ibarra–Krawinkler Relationships

In recent years, the Ibarra–Krawinkler model has been used for better modeling of structural behavior. Among the newer models is the Lignus model, which is based on laboratory observations. Below are some of the modifications made in the model [26]. The conventional approach to relating stiffness after yielding to initial stiffness, stiffness coefficient to, is high and far from expected, and it is better to use the ratio to define the stiffening behavior after the elastic state, which is the value for the beam. Additionally, the columns are obtained between 1.05 and 1.1. In the main model, the ratio is used as the input parameter defining the plastic deformation capacity. Based on the new data, providing better and more direct parameters in determining the behavior of steel components after elastic state and behavior after, respectively, gives a definition of basic energy dissipation capacity in the main model. This definition was considered a multiple of the yield force in the yield deformation. However, in this model, a more stable parameter (plastic deformation) is used [27].

$$E_t = \gamma \times F_Y \times \delta_p \tag{3}$$

It is better to use the parameter $\Delta = \gamma \times \theta_p$ to rotate the plastic joints. Since the parameter $\theta_p$ is dimensionless, the $\Delta$ parameter will be compatible with the relation $\beta_i = \left[ \dfrac{E_i}{E_t - \sum_{j=1}^{i} E_j} \right]^c$, giving a more tangible understanding of plastic rotation. For example, if $\theta_p = 0.5$ and $\gamma = 20$, then $\Delta = 1$.

Figure 9 shows the modified Ibarra–Krawinkler Model used in this study for modeling the connections.

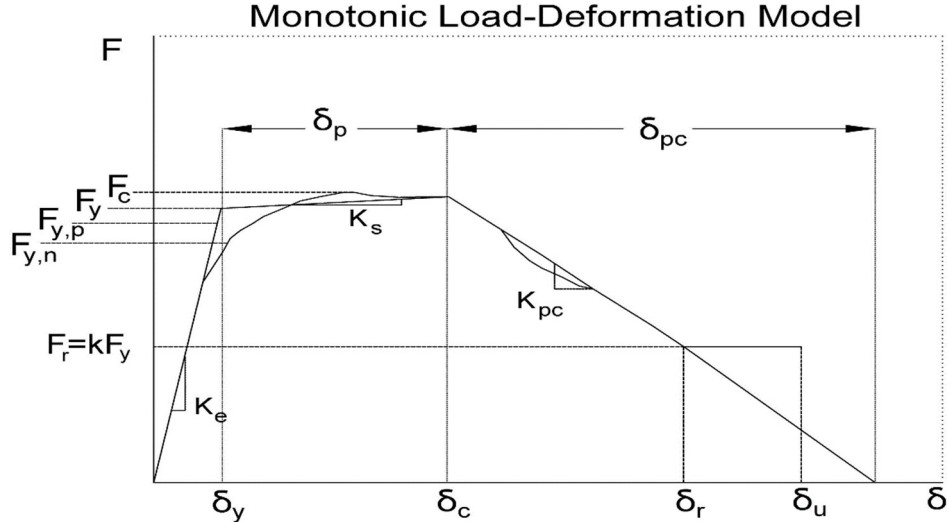

**Figure 9.** Modified Ibarra–Krawinkler Model.

In figure the parameters are:

$\delta_c$: Deformation at the apex point
$F_y$: Effective yield strength (taking into account strain hardening)
$\delta_y$: Effective yield deformation
$K_e$: Effective elastic stiffness
$f_r$: Residual capacity
$\delta_r$: Residual deformation
$\delta_u$: Final deformation capacity
$\delta_p$: Plastic deformation
$\delta_{pc}$: Deformation capacity after Yielding
$F_{yp}$: Predicted yield resistance
$F_{yn}$: Nominal yield resistance
K: Residual ratio

The above model is based on the database collected from cyclic and uniform loading tests on steel members around the world, and after nonlinear regression analysis on the data, the following values are proposed for different sections [27].

(a)    Beam with W-shape section

$$\theta_p = 0.07 \left(\frac{h}{t_w}\right)^{-0.35} \cdot \left(\frac{b_f}{2 \cdot t_f}\right)^{-0.09} \cdot \left(\frac{L}{d}\right)^{0.310} \cdot \left(\frac{d}{c_{unit}^{2.21}}\right)^{-0.281} \cdot \left(\frac{c_{unit}^2}{50}\right)^{-0.383} \tag{4}$$

$$\theta_{pc} = 4.645 \left(\frac{h}{t_w}\right)^{-0.449} \cdot \left(\frac{b_f}{2 \cdot t_f}\right)^{-0.837} \cdot \left(\frac{d}{c_{unit}^2 \cdot 21}\right)^{-0.265} \cdot \left(\frac{c_{unit}^2 \cdot F_y}{50}\right)^{-1.136} \tag{5}$$

$$\Lambda = 26.36 \left(\frac{h}{t_w}\right)^{-0.589} \cdot \left(\frac{b_f}{2 \cdot t_f}\right)^{-0.574} \cdot \left(\frac{c_{unit}^2 \cdot F_y}{50}\right)^{-1.454} \tag{6}$$

(b)    Beam with Box-Shape section

$$\theta_p = 0.572 \cdot \left(\frac{D}{t}\right)^{-1.00} \cdot \left(1 - \frac{N}{N_y}\right)^{1.210} \cdot \left(\frac{c_{unit}^2 \cdot F_y}{50}\right)^{-0.838} \tag{7}$$

$$\theta_{pc} = 14.51 \cdot \left(\frac{D}{t}\right)^{-1.217} \cdot \left(1 - \frac{N}{N_y}\right)^{3.035} \cdot \left(\frac{c_{unit}^2 \cdot F_y}{50}\right)^{-0.498} \tag{8}$$

$$\Lambda = 3800 \cdot \left(\frac{D}{t}\right)^{-2.492} \cdot \left(1 - \frac{N}{N_y}\right)^{3.501} \cdot \left(\frac{c_{unit}^2 \cdot F_y}{50}\right)^{-2.391} \tag{9}$$

In the above relations, $c_{unit}^1$ and $c_{unit}^2$ are for unit conversion. If d is based on millimeters and $F_y$ is based on MPa, $c_{unit}^1 = 25.4$ and $c_{unit}^2 = 0.145$ [21,22].

### 3.2.3. Constitutive Curve

In this method, the nonlinear model of the members is the same as before, but an eight-node model shown in Figure 10 is used for the connection spring. A three-line curve named the constitutive curve shown in Figure 11 is used to consider the nonlinear behavior in the model. In this model, the dimensions of the beam and column are considered the dimensions of the connection spring. To assign a three-line behavior, a three-line torsion spring or two two-line torsion springs can be used in one of the corners. The first slope after surrender is steeper, which indicates the behavior of the connection spring after surrender until the maximum resistance is reached. After reaching the maximum slope capacity, less than 0.2 can be reasonable. In this model, it is assumed that the connection spring reaches four times the full plastic capacity in the deformation of the yield [26,28].

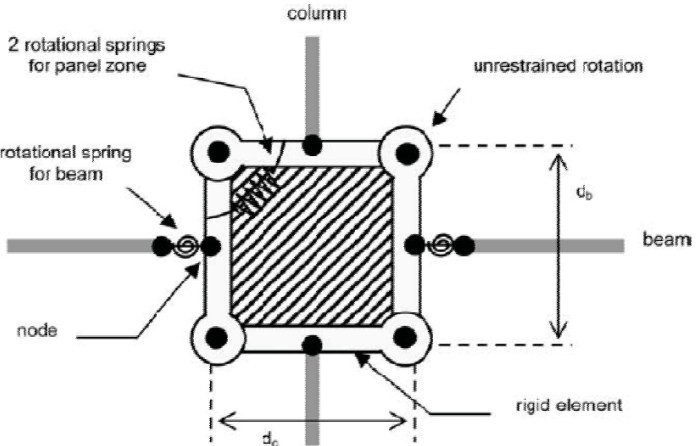

**Figure 10.** Connection source model using an eight-node model.

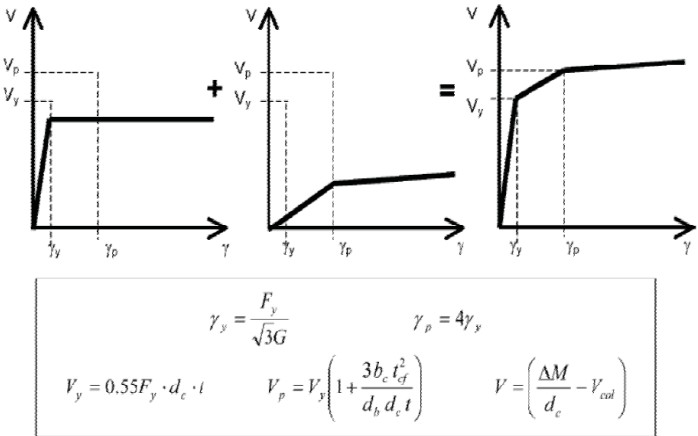

**Figure 11.** Force behavior displacement of the connection source.

### 3.2.4. Modeling of Beams and Columns Using Ibarra Model

To define the sections, fiber filament sections that have the ability to consider the interaction of axial and flexural forces, or non-linear beam-column elements with wide plasticity, the analysis of which is based on the method of force or displacement, can be used. The above methods are advanced methods but are time consuming and require

accuracy to converge in the analysis. Considering this, in this dissertation, the purpose of modeling is to study the behavior of structures near demolition surfaces, and in these levels the effect of stiffness and strength is very significant, so the most appropriate way to model these effects accurately is to use concentrated plasticity [29]. For this purpose, an elastic beam with zero plastic hinge at both ends is used. The behavior of plastic connections at both ends of the member is defined by considering the stiffness and strength deterioration in the two-line model and is modeled using the modified Ibarra-Krowinkler model in OpenSees software. However, due to programming problems in OpenSees software to reduce the difficulty of loading, the effect of this part has been omitted. In this case, the flexural stiffness of the member can be obtained based on the specifications of the molds, which is considered to be 6EI/L for beams and 3EI/L for columns. The rotational stiffness of the member is directly related to the torsional spring stiffness of $K_s$ and the inelastic stiffness of $K_{BC}$, which is obtained from the following equation:

$$K_{mem} = \frac{1}{\frac{1}{K_s} + \frac{1}{K_{BC}}} = \frac{K_s \times K_{BC}}{K_s + K_{BC}} \tag{10}$$

There are three methods for assigning the stiffness of columns and beams.

$$K_{BC} = \infty, \ K_s = \ K_{mem} \tag{11}$$

$$K_{BC} = \ K_{mem}, \ K_s = \infty \tag{12}$$

$$K_{BC} = \ K_s = 2 \times \ K_{mem} \tag{13}$$

In the above relationships, $K_{mem}$ is equivalent to member stiffness. In this dissertation, the third method is used, and therefore the parameters related to the two-line model should be modified as follows:

For strain stiffness and the stiffness of a curved branch with a negative slope:

$$a_{ss} = \frac{a_{smem}}{n + 1 - n \times a_s} \tag{14}$$

To modify the relative "displacement" to the second branch displacement (to modify the cyclic reduction parameter):

$$\left(\frac{\delta_c}{\delta_y}\right)_S = \left(\left(\frac{\delta_c}{\delta_y}\right) - 1\right)(1 - a_{smem}) \times n + \left(\frac{\delta_c}{\delta_y}\right)_{mem} \tag{15}$$

$$a_s = (n + 1)a_m \tag{16}$$

There are three ways to convert local coordinates to general coordinates in OpenSees Ver. 2.5 software. The first method, which is a linear method, ignores the effect of large deformations and P-Delta. The second method is the P-Delta method, which is used for columns in this dissertation. Co-rotational change of coordinates has also been used in modeling braces. When building a model with high degrees of freedom in OpenSees software, "usually" occurs in solving numerical instability equations, the main reason being how the mass is assigned to the degrees of freedom in the structure. If the ceiling mass is spread between a small number of degrees of freedom, the mass matrix will be a matrix with low numerical density, and this phenomenon will cause numerical instability during nonlinear analyses, so in order to prevent this, small amounts of mass can be assigned to different degrees of freedom in the structure. For this purpose, the main mass was applied at the top of the connection and small masses (equal to 0.005% of the main mass) were assigned to other degrees of transfer and rotational freedom. Moreover, considering that there is a possibility of elevation in the column plate in braced structures, the effect of this elevation should be seen in the modeling. For this purpose, a zero-length spring is used between the column and the foundation, so that the stiffness of this spring is very high in

the direction in which the column is pressed, and in the direction where the column works in tension, which means zero spring stiffness [26,30].

### 3.2.5. Ground Motion Records

The first step in the process of evaluating the performance of IDA curves is to prepare a set of earthquake ground motions, which indicate the seismicity of the area. In fact, if a sufficient number of seismic accelerometers have been recorded in the area in question, those will be used; otherwise, similar ones from the PEER website can be utilised. Therefore, in the first step, records should be selected in accordance with the almost similar conditions of the region in terms of fault mechanism, distance from the desired site and the magnitude of the earthquake. Due to the shortcomings in the accelerometers recorded in Iran, and ambiguities being ambiguities in their accuracy, in order to reduce the number of errors in this study, a selection of the selection of accelerometers from 15 accelerometers of the recommended records FEMA-P695 [31] has been used, as mentioned in the reference. The specifications of these accelerometers are also summarized in the table below. It should be noted that, in this paper, each of these 15 accelerometers has been scaled according to the spectral acceleration of each at 5% damping and the period mode cycle of the structure Sa (0.05, T1). For this purpose, Seismosignal and Excel software have been used in this article. In this way, first, the spectral acceleration of each record is brought to 1 and then according to the modeling and analysis file of each system, each of the selected records is from 0.1 g to the limit of structural failure (which here is approximately 1.5 g). Table 3 shows the specifications of the used ground motions. Figures 12 and 13 show the large-distance distance diagram and the average response spectrum of selected ground motions, respectively.

**Table 3.** Selected used ground motions.

| Record Number | Earthquake Name | Station | Year | Magnitude | Distance | Soil Type | Fault Type | Maximum PGA |
|---|---|---|---|---|---|---|---|---|
| 1 | Northridge | Beverly Hills - Mulhol-USC | 1994 | 6.7 | 17.2 | D | Thrust | 0.52 |
| 2 | Northridge | Canyon Country-WLC-USC | 1994 | 6.7 | 12.4 | D | Thrust | 0.48 |
| 3 | Duzce, Turkey | Bolu-ERD | 1999 | 7.1 | 12 | D | Strike-slip | 0.82 |
| 4 | Chi Chi | Chi Chi | 1999 | 7.7 | 8.1 | C | Strike-slip | 0.34 |
| 5 | Imperial Valley | Delta-ENAMUCSD | 1979 | 6.5 | 22 | D | Strike-slip | 0.35 |
| 6 | Imperial Valley | El Centro Array #11-USGS | 1979 | 6.5 | 12.5 | D | Strike-slip | 0.38 |
| 7 | Kobe, Japan | Nishi-Akashi-CUE | 1995 | 6.9 | 7.1 | C | Strike-slip | 0.51 |
| 8 | Kobe, Japan | Shin-Osaka-CUE | 1995 | 6.9 | 19.2 | D | Strike-slip | 0.24 |
| 9 | Bam | Bam | 2003 | 6.6 | 8.5 | D | Reverse | 0.36 |
| 10 | Tabas | Tabas | 1978 | 7.7 | 10 | C | Reverse | 0.2 |
| 11 | Kocaeli, Turkey | Duzce-ERD | 1999 | 7.5 | 15.4 | D | Strike-slip | 0.36 |
| 12 | Kocaeli, Turkey | Arcelik-KOERI | 1999 | 7.5 | 13.5 | C | Strike-slip | 0.22 |
| 13 | Manjil, Iran | Abbar-BHRC | 1990 | 7.4 | 12.6 | C | Strike-slip | 0.51 |
| 14 | Loma Prieta | Capitola-CDMG | 1989 | 6.9 | 15.2 | D | Strike-slip | 0.53 |
| 15 | Loma Prieta | Gilroy Array #3-CDMG | 1989 | 6.9 | 12.8 | D | Strike-slip | 0.56 |

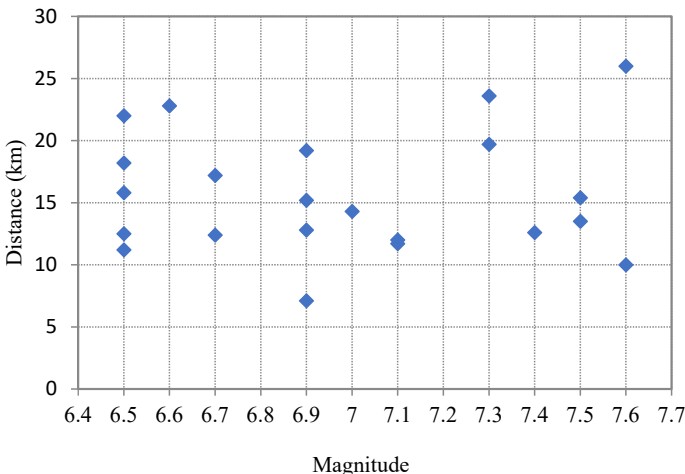

**Figure 12.** Large-distance distance diagram for selected ground motions.

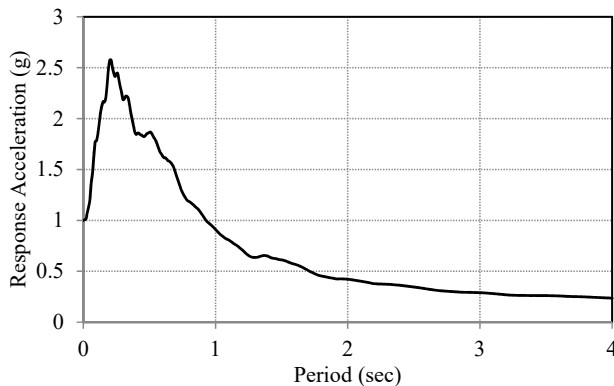

**Figure 13.** The average response spectrum of selected ground motions.

3.2.6. Results and Discussions

As mentioned, the selected frames of the analyzed structures were modeled in two dimensions in OpenSees software and the general outline of the modeled frames is given in the following figures. The response of an earthquake structure can be estimated with appropriate accuracy by performing dynamic analysis of time history. One of the most important drawbacks of applying nonlinear dynamic analysis is the sensitivity of the response to selected accelerometers. The presentation of incremental dynamic analysis and estimation of responses based on the application of probabilistic relations has to a large extent compensated for this weakness in practice. The results of IDA analysis and comparison of their means of 50% for all frames are shown in Figures 14–21.

As can be seen from the figure above, the cover curve for the 15-story structure with semi-rigid connections is the lowest and the curve of the 3-story structure with rigid connections is at the highest level, which indicates that the corresponding curve curves. Structures with less ductility are placed at a higher level, in other words, with increasing ductility in nonlinear static analysis (coating), and the coating curve is placed at a lower level. In other words, with increasing stiffness of connections, the structure has a stiffer behavior and as a result, their curvature is at a higher level.

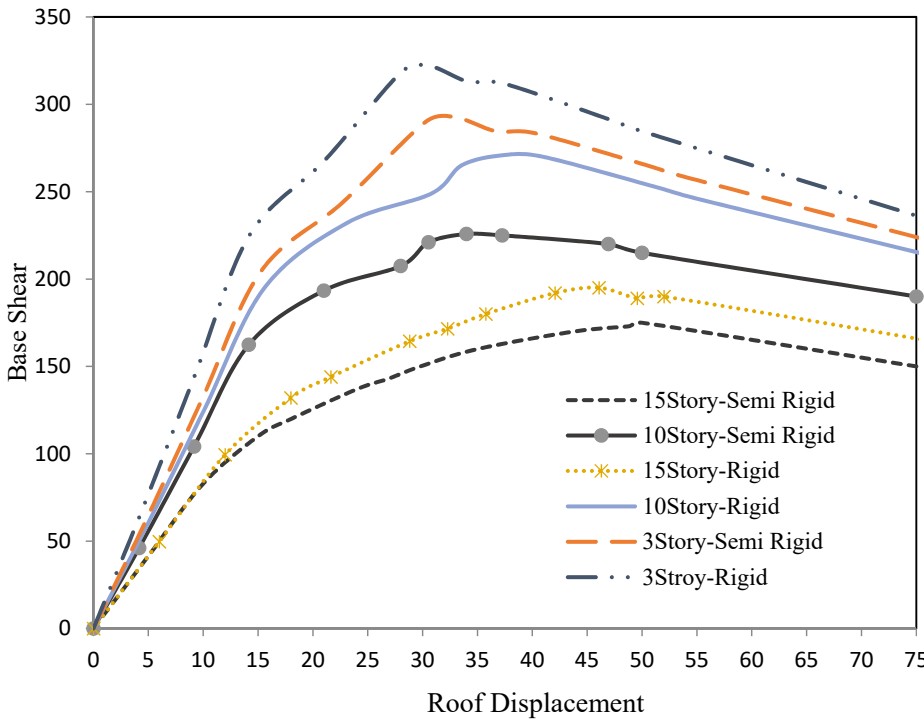

**Figure 14.** Nonlinear static curves.

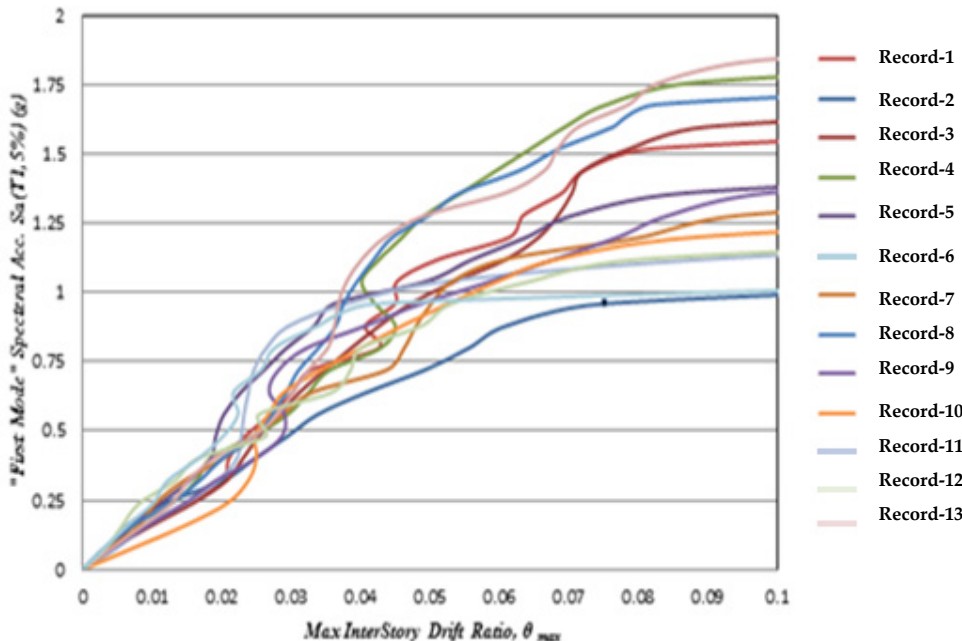

**Figure 15.** IDA curves obtained for a 15-story building with simple connections.

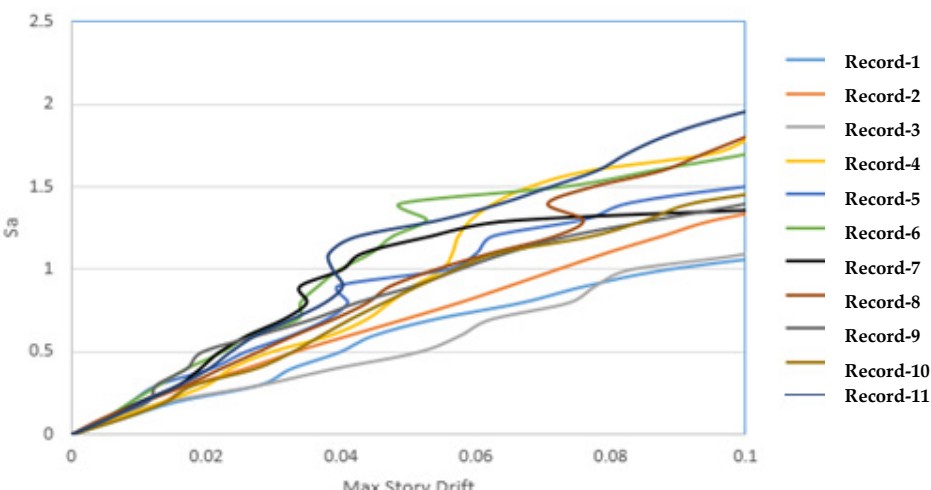

**Figure 16.** IDA curves obtained for a 15-story building with semi-rigid connections.

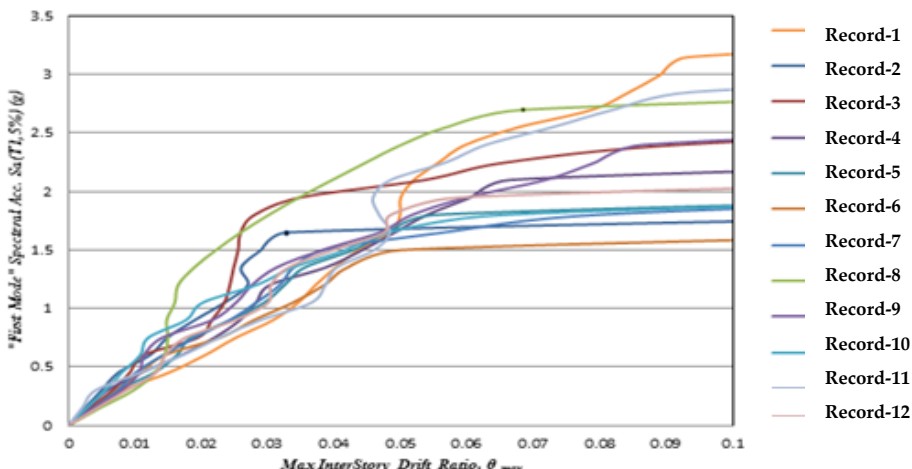

**Figure 17.** IDA curves obtained for a 10-story building with simple connections.

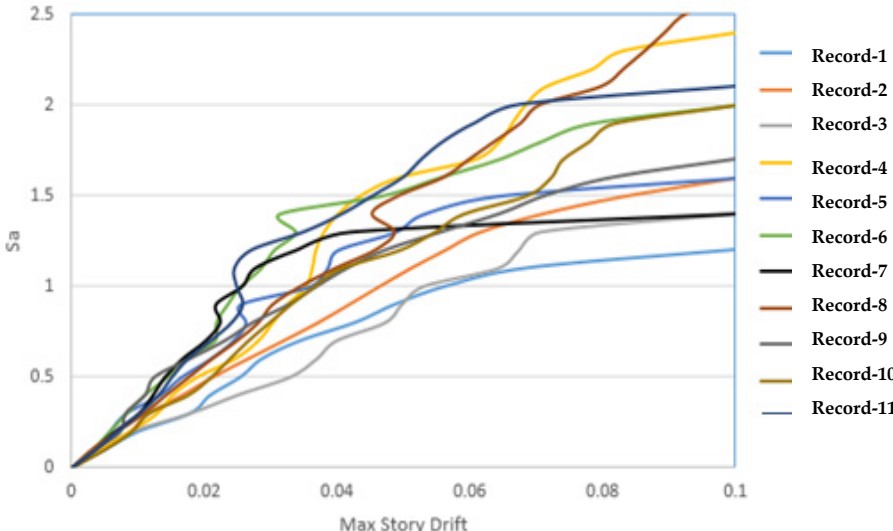

**Figure 18.** IDA curves obtained for a 10-story building with semi-rigid connections.

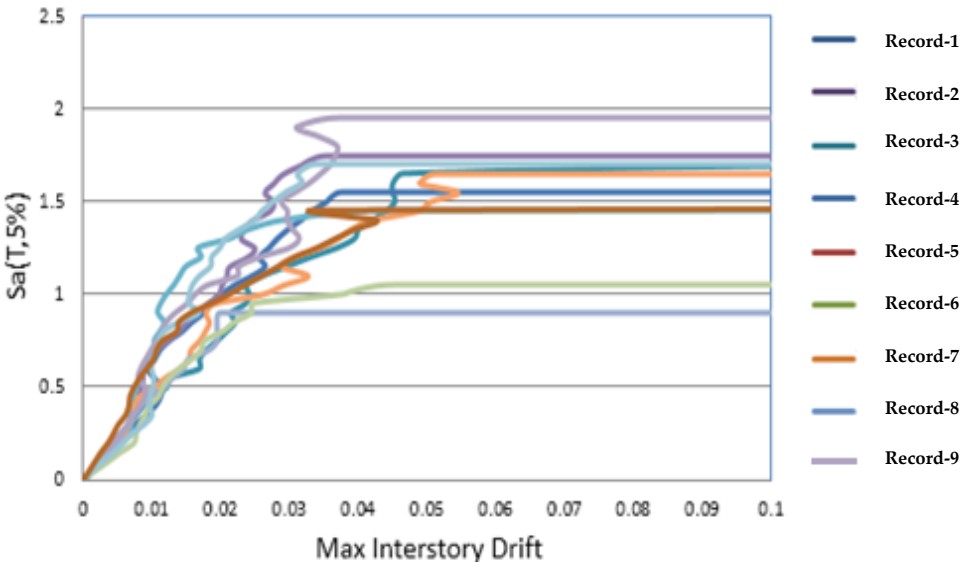

**Figure 19.** IDA curves obtained for a 3-story building with simple connections.

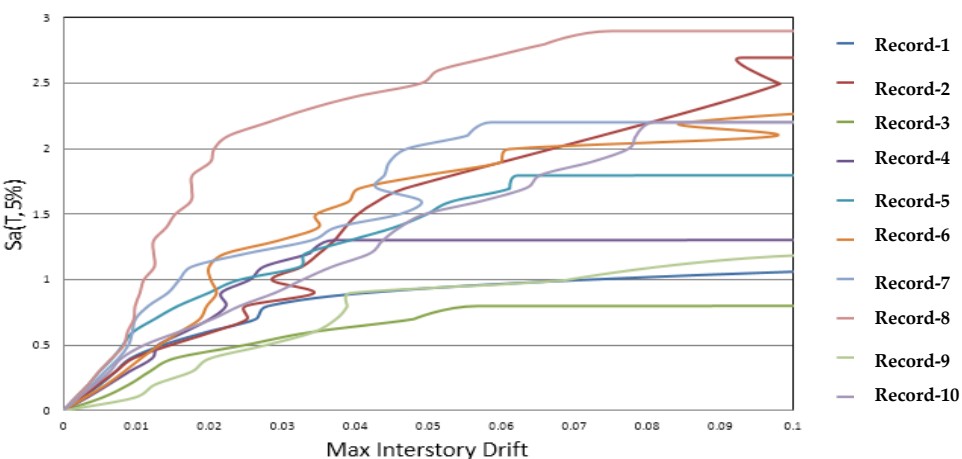

**Figure 20.** IDA curves obtained for a 3-story building with semi-rigid connections.

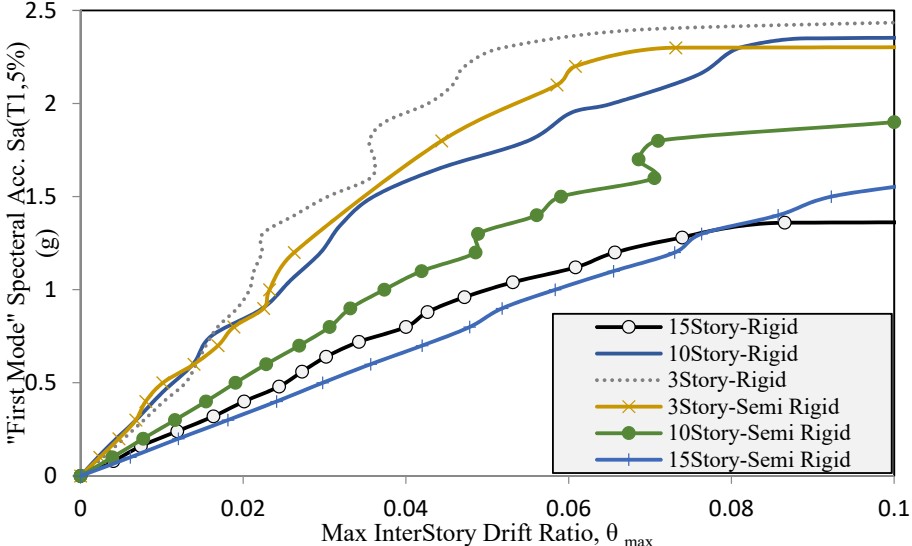

**Figure 21.** Comparison of 50% curves for IDA curves obtained for all cases.

As observed, the results of incremental nonlinear dynamic analysis (IDA) were seen in the above figures for the 15 accelerometers introduced. To compare the results, it is recommended that these curves be summarized and then compared with each other. In the figure above, a comparison of the average of 50% of the curves for all six structures is performed. As can be seen from the figure above, the more ductile structures become horizontal at a lower seismic intensity level and the stiffer structures are placed at a higher level, which is the case in the results of nonlinear static analysis. Has also appeared. In other words, the building as a complex system consists of structural and non-structural members that are connected by different types of connections. Given the importance of connections, accurate knowledge of their behavior, especially during an earthquake, is essential. In the past, according to the level of knowledge of structural analysis, simple modeling was attempted in the building, and thus the connections were divided into two types: completely rigid (clamped) and fully flexible connections. However, today it is well known that every rigid connection has a certain amount of flexibility, and every connection can withstand a certain amount of anchor, which makes the semi-rigidity of the connections quite clear. Extensive local damage to rigid welded connections due to the 1994 Northridge earthquake led to numerous studies on the ductility of welded connections in areas with severe seismicity. One of the suggestions in this regard is the use of semi-rigid connections; in severe earthquakes, on the one hand, this connection is effective in the dissipation of earthquake energy through proper rotational ductility, and on the other hand, this connection with increasing damping is effective in reducing the force of earthquakes. In the study of semi-rigid connections, it is necessary to determine the strength and ductility of the connection, and thus after performing linear and non-linear analyses under the influence of gravity and lateral loads, unexpected failures and uncertain behaviors of the connections should be prevented. According to the research, paying attention to the semi-rigidity of the connections, in addition to making the connections less economically expensive, helps to identify the exact behavior of the connections and leads to economic optimization and seismicity of steel structures. In Iran, due to the lack of high-profile profiles for beams in medium and short buildings, two sections of beams with a distance between them are used. In the semi-rigid type of connections, this study is passed along the columns due to the lack of sufficient space for connection at the location of the beams. Fast construction and installation, cutting and less welding, as well as more economical design, are benefits compared to beams with simple support due to their uniformity and less deformation under earthquake load and the ability to be used in temporary structures and the possibility of reusing profiles. It can be one of the advantages of using this type of semi-rigid connections in the country. Of course, the rotational nature of this type of connection is somewhat unknown. Therefore, due to the semi-rigid state, some anchor is transferred to the column beam and the connection must be capable of this transfer. Moreover, due to the semi-rigid connection, the whole frame must be inspected with a semi-rigid connection and considering the criteria of frames. The semi-rigid connection is analyzed.

### 3.2.7. Comparison with the Previous Studies

In order to validate this article, we can refer to the work conducted in the dissertation and the second extraction article with the guidance of Banazadeh at Amirkabir University in 2013 entitled "Decision analysis for seismic improvement of conventional steel structures based on Risk management and performance-based design" [24]. They first analyzed and designed the structures in 3D-SAP software and then extracted a frame representing the whole structure (critical frame) and analyzed nonlinear dynamic analysis in OpenSees software. They also selected several accelerometers and performed scaling, obtaining IDA curves and fragility curves. Of course, they have advanced to the stage of determining the damage, which was not the starting point of this research for high-rise structures. Their results can also be seen in the following figures. As can be seen in Figure 22, there is a good match between the results and the reference article [32].

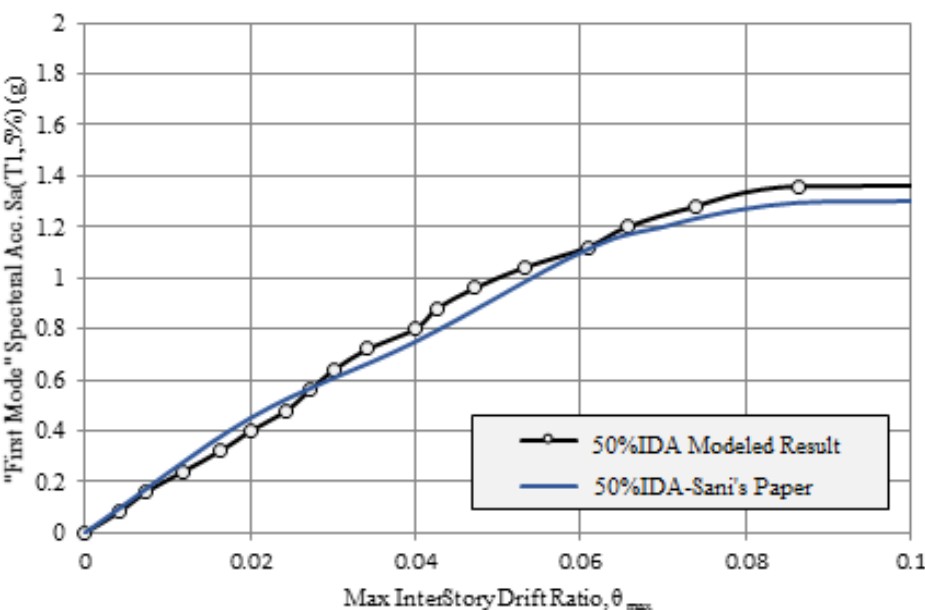

**Figure 22.** Comparison between 50% IDA curves.

## 4. Performance-Based Analysis

Given the performance-based approach to earthquake engineering, as well as the importance and necessity of proper and principled management of potential crises such as earthquakes, there is a need for tools for the real-world assessment of structural and non-structural systems as well as proper damage estimation. Possible and subsequent rational decisions become more and more important and necessary. Structural assurance has always been a principle for engineers responsible for designing construction projects. One of the mechanisms that has received increasing attention in recent decades is called progressive failure, in which one or more members of a structure suddenly collapse due to an accident or attack, after which the building progressively collapses. What will be discussed in this section is a detailed study of these types of failure in steel structures, which has so far been less addressed in Iran. In this regard, we will study this type of structure, which is widely used in metropolises such as Tehran today. What is expected to be received from this paper is the behavior of the structure in two failure modes that occur progressively in the structure in question, so that the results of the failure curves in this paper in three stages (a healthy mode, and two modes relating to the removal of several beams and columns) have been compared with each other. Therefore, the failure mode of the structure will be considered a variable in this structure. In this study, three modes have been considered, and the results have been compared with each other.

In order to achieve the goals of this project, we first model a 20-story steel structure with a three-dimensional cross brace (Figure 23 and Table 4), which is assumed to be located in a high seismic hazard zone on Type IV soil, in Etabs 2016 Ver. 11 software. The reason for selecting this number of floors is to review and evaluate a high-rise structure during failure, and for this reason, a structure with more than 15 floors has been selected. Then, we design this structure based on the topics of steel (No. 10) and concrete (No. 9) edition 1392 and seismic guide 2800 edition 4 (the latest edition of national building regulations). In this way, we design the necessary sections for columns and beams. Then, we select a side frame of this structure that seems to be a more critical frame and we will use it in nonlinear software and perform nonlinear analysis on them. In the next step, considering that in this project the aim is to investigate the progressive damage in this structure, it is assumed that due to an explosion or a special impact that occurs on the fifth floor of this structure, a beam and a column on the side frame of this structure lost its destruction, load and transmission of force and is thus removed from the model. In the next step, it is assumed that a column

and another beam on the tenth floor of this structure will be damaged in addition to the previous state and will be out of the circle of structural elements.

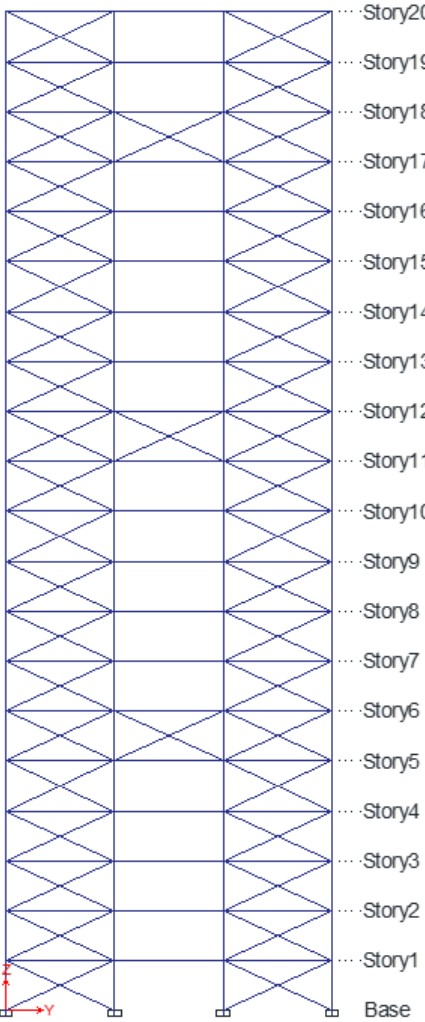

**Figure 23.** Schematic view of the side 20-story frame in ETABS.

**Table 4.** Sections used in structural frames selected from existing buildings. Modes have been considered and the results have been compared with each other.

| 20Story Building | | | |
|---|---|---|---|
| **Story** | **Columns** | **Beams** | **Braces** |
| 1–4 | Box 600 × 600 × 40 | IPE 400 | Box 160 × 160 × 16 |
| 5 | Box 450 × 450 × 35 | IPE 400 | Box 160 × 160 × 16 |
| 6 | Box 450 × 450 × 35 | IPE 400 | Box 180 × 180 × 20 |
| 7–11 | Box 450 × 450 × 35 | IPE 360 | Box 160 × 160 × 16 |
| 12 | Box 450 × 450 × 35 | IPE 360 | Box 180 × 180 × 20 |
| 13–17 | Box 350 × 350 × 25 | IPE 330 | Box 140 × 140 × 10 |
| 18 | Box 350 × 350 × 25 | IPE 330 | Box 180 × 180 × 20 |
| 19–20 | Box 350 × 350 × 25 | IPE 330 | Box 140 × 140 × 10 |

In this way, the behavior of the structure due to progressive failure (equivalent to the mentioned case) can be observed and the probability of failure for each step can be

examined. In fact, in this project, three states of healthy structure and damaged structure on the fifth floor (step 1) and damaged structure on the fifth and tenth floor (step 2) were observed and compared with each other in the face of various earthquakes. In the final step, nonlinear analysis is performed for each of these three modes and IDA curves and fragility curves are obtained for all three modes. In the software, increasing nonlinear dynamic analysis is performed on these frames under selective earthquake records. In this study, the relative displacement of structures is considered as a criterion and indicator of damage. In each analysis, the desired failure levels are determined. After performing nonlinear dynamic analyses for selected earthquake records, the maximum ground acceleration versus the maximum relative drift of the floors has been determined, and finally the fragility curves of the structures for the collapse level (CP) functional level will be obtained. An overview of the structures modeled in the software can be seen in Figures 24–32.

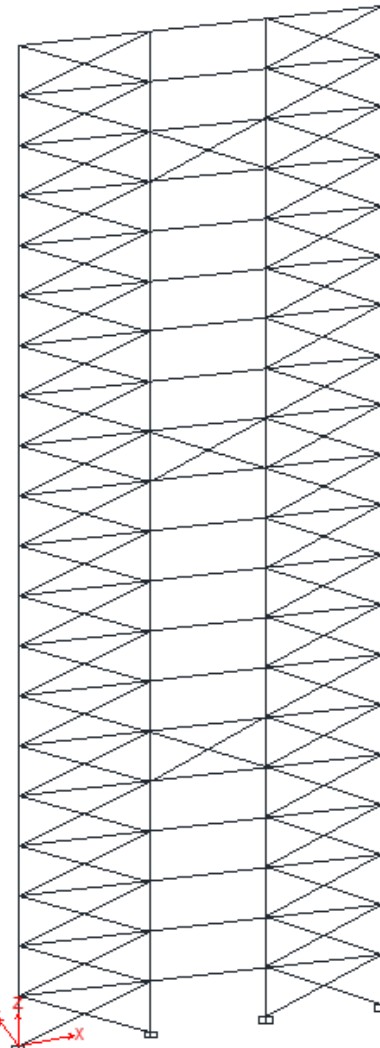

**Figure 24.** Schematic view of the 20-story model of an undamaged frame in Seismostruct.

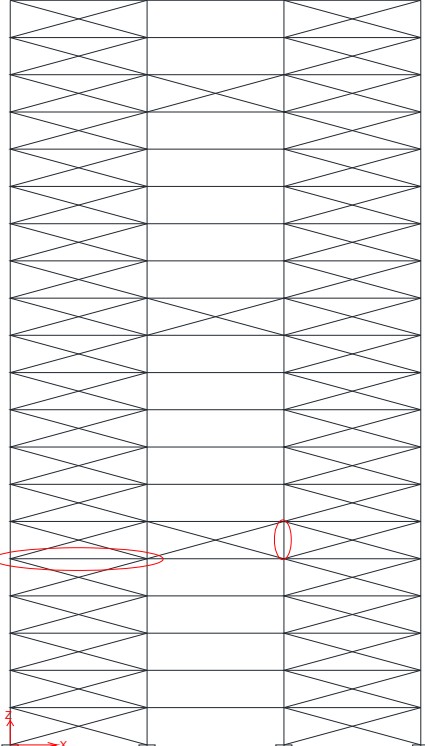

**Figure 25.** Schematic view of Step 2 damaged 20-story frame (by removing a beam and a column on the fifth floor). The removed beam and column are marked in the figure with a red circle.

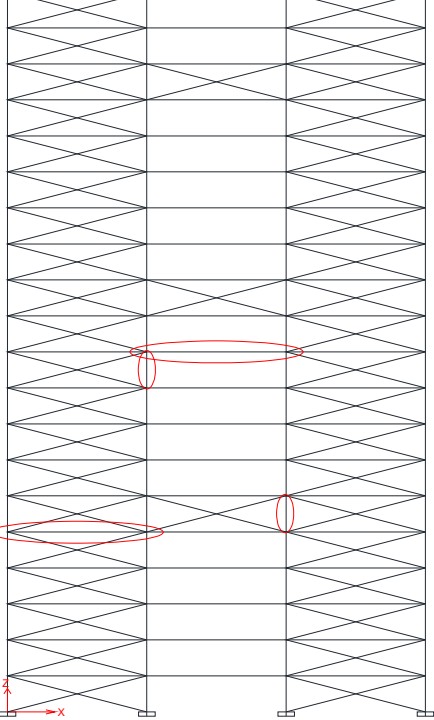

**Figure 26.** Schematic view of the Step 2 damaged 20-story frame (by removing one beam and one column on the tenth floor in addition to removing one beam and one column on the fifth floor). The beam and column removed are shown with a circle marked in red.

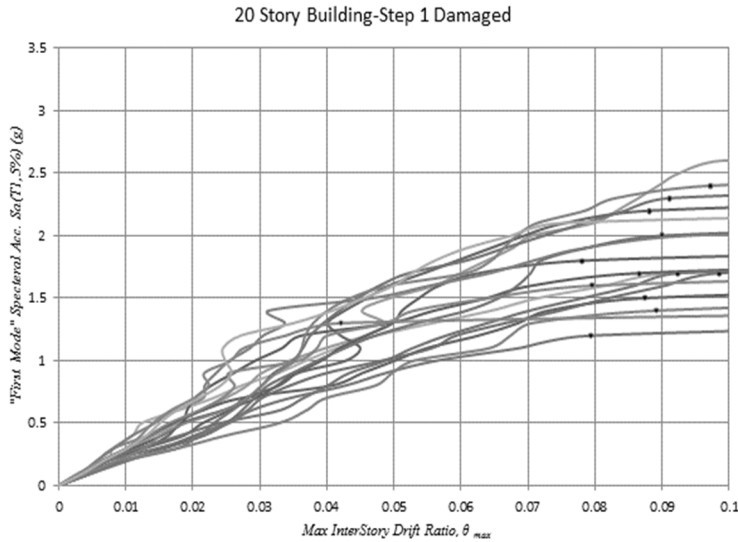

**Figure 27.** IDA curves obtained for a slightly damaged 20-story frame (by removing a beam and a column on the fifth floor).

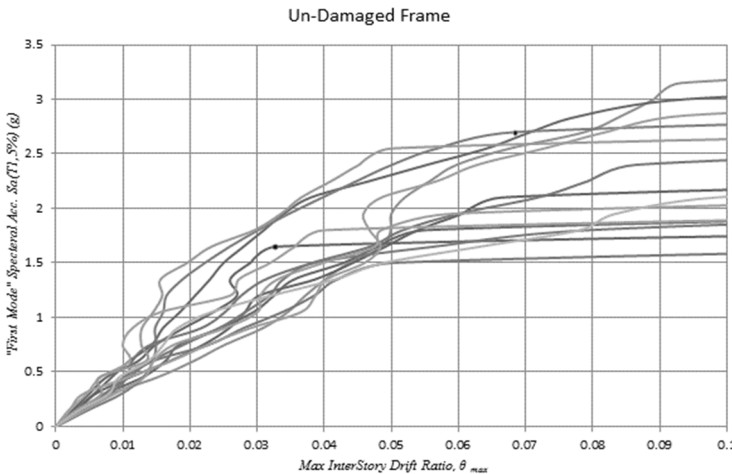

**Figure 28.** IDA curves obtained for a sound 20-story structural frame.

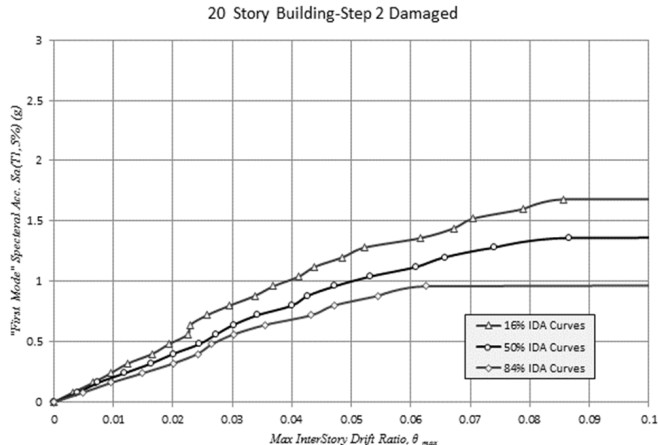

**Figure 29.** Curves of 16%, 50%, and 84% for IDA curves obtained for a slightly damaged 20-story frame (by removing one beam and one column on the tenth floor in addition to removing one beam and one column on the fifth floor).

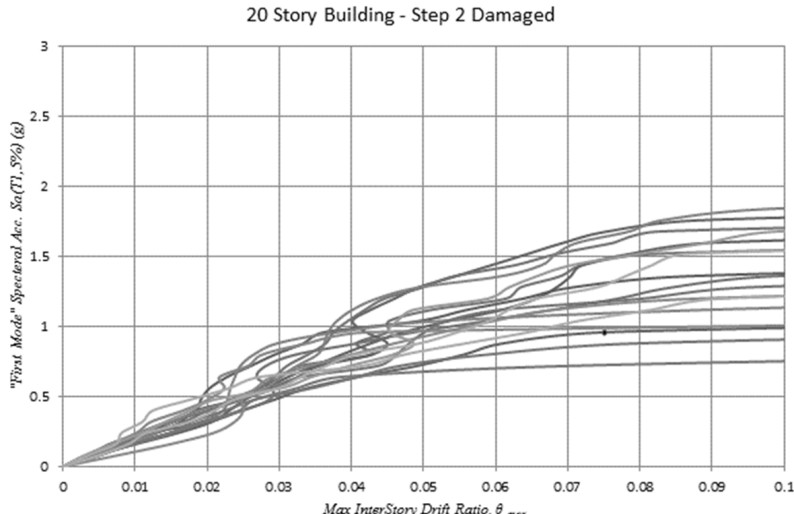

**Figure 30.** IDA curves obtained for a slightly damaged 20-story frame (by removing one beam and one column on the tenth floor in addition to removing one beam and one column on the fifth floor).

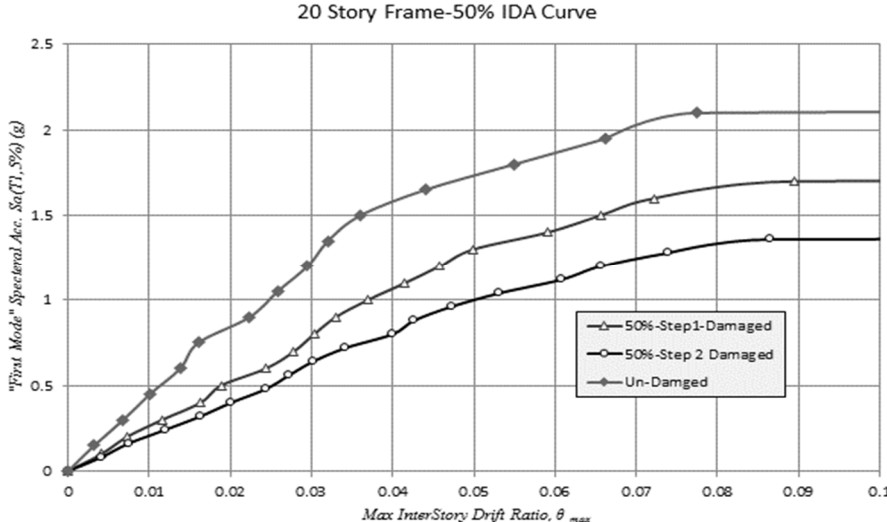

**Figure 31.** Comparison of 50% curves of IDA curves obtained for all 3 structural frames.

The seismic hazard diagrams (Table 5) used in this part of the study are adapted from the studies of Mahdavi Adeli and Banazadeh. In the study, based on the various reduction relations available, seismic hazard in different parts of Tehran has been estimated, and by averaging the seismic hazard values obtained, a uniform hazard map of Tehran has been obtained. Seismic hazard graphs, which express the average annual incidence rate of passing different seismic intensity values, have been obtained using a uniform hazard map for different structural periods. Moreover, MAF transmittance of seismic intensity in terms of Sa is estimated by a linear relationship in the log-log space. The relation related to this estimate is as follows [15].

$$\lambda_{S_a} = k(S_a)^t \tag{17}$$

The parameters k and t adapted from the study for the fluctuation periods related to the buildings discussed in this study for the high-risk area are given in the Table 6. The uniform hazard curves obtained for all three frames are shown in the Figure 33.

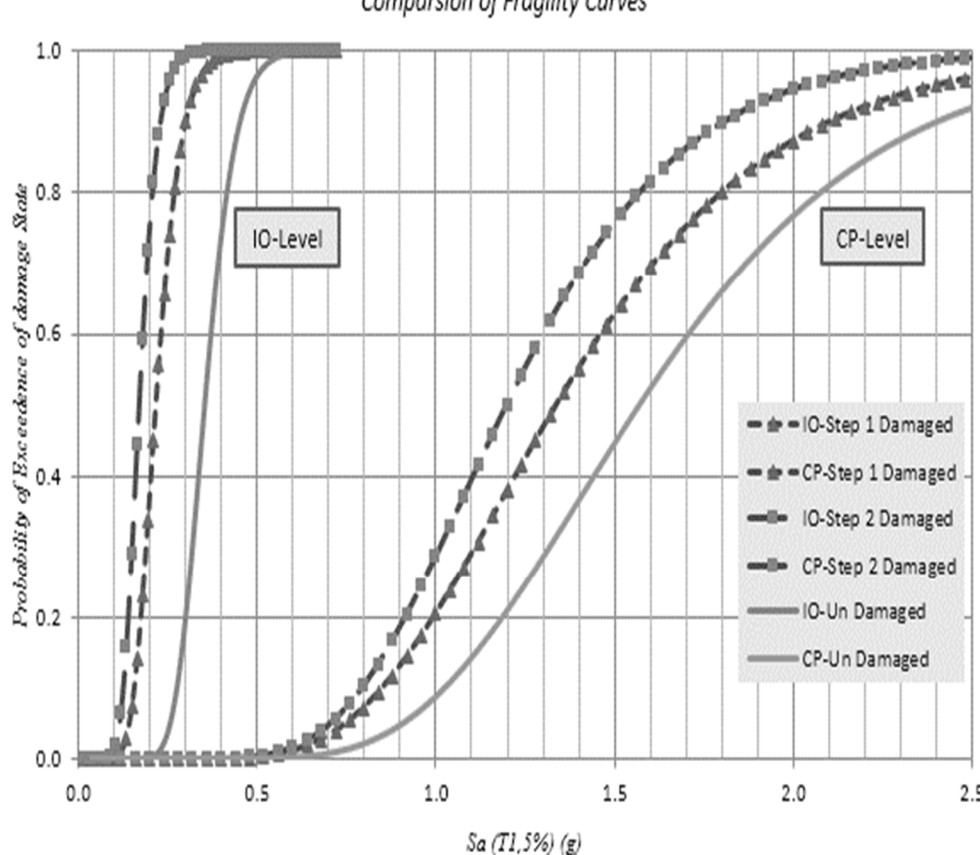

**Figure 32.** Comparison of fragility curves obtained from IDA curves for the level of failure corresponding to the level of failure of CP and IO for all three structural frames.

**Table 5.** Parameters in seismic hazard curves for three different zones.

| Spectral Acceleration | High Level Hazard | | Medium Level Hazard | | Low Level Hazard | |
|---|---|---|---|---|---|---|
| | k | t | k | t | k | t |
| $S_a$ (0.30) | $1.890 \times 10^{-3}$ | $-2.653$ | $8.422 \times 10^{-4}$ | $-2.683$ | $1.861 \times 10^{-4}$ | $-2.888$ |
| $S_a$ (0.60) | $5.653 \times 10^{-4}$ | $-2.131$ | $2.661 \times 10^{-4}$ | $-2.191$ | $1.861 \times 10^{-5}$ | $-2.510$ |
| $S_a$ (0.90) | $1.787 \times 10^{-4}$ | $-2.005$ | $8.947 \times 10^{-5}$ | $-2.105$ | $1.861 \times 10^{-5}$ | $-2.367$ |
| $S_a$ (1.20) | $7.460 \times 10^{-5}$ | $-2.021$ | $3.444 \times 10^{-5}$ | $-2.140$ | $1.861 \times 10^{-6}$ | $-2.451$ |
| $S_a$ (1.50) | $5.356 \times 10^{-5}$ | $-2.021$ | $2.473 \times 10^{-5}$ | $-2.140$ | $1.861 \times 10^{-6}$ | $-2.451$ |

**Table 6.** Calculated parameters k and t used in Equation (17).

| t | k | $T_1$ | |
|---|---|---|---|
| $-1.98$ | 0.000137 | 0.965 | 20 story frame-slightly damaged (by removing one beam and one column on the tenth floor in addition to removing one beam and one column on the fifth floor) |
| $-2.03$ | 0.000351 | 0.701 | 20-story frame slightly damaged (by removing a beam and a column on the fifth floor) |
| $-2.24$ | 0.000159 | 0.568 | Structural frame of 20 undamaged frames |

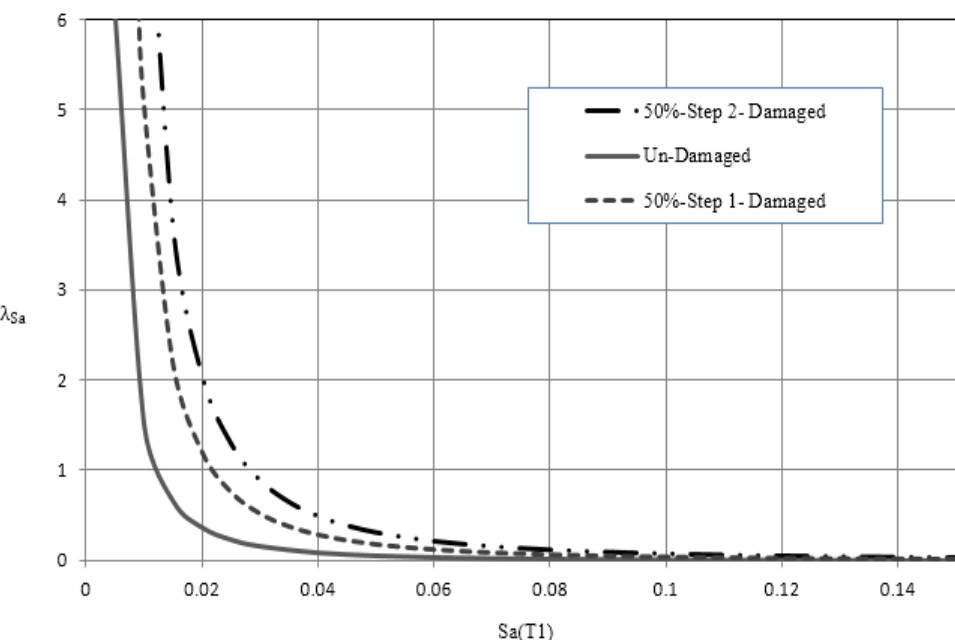

**Figure 33.** Comparison of uniform hazard curves for three frames.

In order to calculate the Mean Annual Frequency of limit states (MAF) in this study, these values are very useful as quantities that reflect the probabilistic capacity of all structures based on earthquake uncertainties. These values can be used as a criterion for measuring the structural reliability of the buildings in comparison with other structures, or they can be used in the regulations related to the design of buildings [1]. In order to calculate the mean annual frequency, the following equation has been used.

$$\text{(Collapse)} = \int_0^\infty \text{P}[\text{Collapse}|\text{IM} = \text{im}_i] \left| \frac{d\lambda(\text{IM} > \text{im}_i)}{\text{dim}} \right| d(\text{im}) \tag{18}$$

In this equation (Collapse), $\lambda$ represents the functions of the mean annual incidence rate (MAF) of transmittance for IM, in which the quantity within the absolute value, the risk gradient of IM and the probability of failure or the same value is a function of fragility. Using high seismic hazard diagrams and fragility and numerical integration diagrams, Equation (18) is used. The values related to the mean annual incidence rate (MAF) of the IO and CP limit states for the structures are shown in the Table 7. These values are very useful as quantities that reflect the probabilistic capacity of the entire structure based on earthquake uncertainties. These values can be used as a criterion for measuring the structural reliability of the buildings in comparison with other structures or they can be used in the regulations related to the design of buildings [16].

**Table 7.** Mean annual frequency for three frames.

| IO-LEVEL | CP-LEVEL | |
|---|---|---|
| $6.1 \times 10^{-4}$ | $1.17 \times 10^{-5}$ | 20 story frame somewhat damaged (by removing one beam and one column on the tenth floor in addition to removing one beam and one column on the fifth floor) |
| $3.6 \times 10^{-4}$ | $1.11 \times 10^{-4}$ | 20 story frame slightly damaged (by removing a beam and a column on the fifth floor) |
| $2.43 \times 10^{-4}$ | $7.49 \times 10^{-5}$ | Structural frame of 20 floors |

The results show that with decreasing the ductility of structures, the mean annual frequency of limit states (MAF) decreases at the functional level of IO, and this is the opposite at the functional level of CP.

## 5. Conclusions

The main findings of the present study are concluded as follows:

(1) The average amount of the ductility reduction coefficient in structures without RBS obtained 1.06 times that of structures with RBS connection. Therefore, energy dissipation capacity in structures with RBS connection is higher than in structures without RBS connection.

(2) Local analysis of connections showed a 9% increase in plastic rotation capacity if RBS connections are used. However, the increase in the coefficient of behavior of steel frames modeled with such a connection was not significant. It seems that connecting the beam with the reduced cross-section of the bending connection beams, by concentrating the stresses in a place away from the connection, is a suitable solution to solve this problem.

(3) The ductility of all frames with RBS connection was increased compared to frames without RBS. The increasing amount was about 0.03% and 0.11 in terms of the behavior coefficient and the ductility coefficient, respectively.

(4) Regardless of the seismic hazard values in calculating the failure probability, with decreasing the rotation time, the probability of collapse (or not estimating the CP functional level) in the structure decreases at a constant level of seismic intensity.

(5) The ductility of structures decreased with decreasing the number of structural layers. The curves obtained by incremental nonlinear dynamic analysis (IDA) became horizontal at a lower seismic intensity level, which indicates that with increasing ductility of structures the higher drifts can be achieved in structures at the same seismic level, which witnessed better behavior of the structures.

(6) As the stiffness of the connections decreases, the effective period of the structure increases and the spectral acceleration decreases accordingly.

(7) The results of folding curves showed that the use of RBS semi-rigid connections has a significant effect on the linear part of IDA curves and with increasing drifts, the reduction in structural resistance against seismic forces is significant.

(8) The RBS semi-rigid connection increases the demand for ductility in beams. In other words, the results of the nonlinear analysis showed that the increase in the behavior coefficient in the connection with the beam with reduced life is more than the normal connection.

(9) With decreasing connection stiffness, the percentage of participation of higher modes has increased, so in structures with semi-rigid connections, the number of effective modes in the structure is higher than the rigid state ratio.

(10) By reducing the stiffness of the connections, the ductility of the structures increases and also the displacement of the structures increases.

(11) As the stiffness of the connections increases, the amount of knot rotation decreases and the final strength also decreases.

(12) The studied structures with high ductility are probably less damaged annually than the other cases, and the structure with the shape Low acceptability can be considered as a next option.

(13) By using the results and the methods introduced in this study, the damage curves can be easily obtained for the country of IRAN, which is part of the ATC58 project to determine the possible damage to structures and its application in the country's insurance industry.

**Author Contributions:** Conceptualization, M.S.; Data curation, M.M.; Investigation, M.M.; Methodology, M.M. and M.S.; Project administration, M.S.; Resources, M.M.; Software, M.M.; Supervision, M.S.; Validation, M.S.; Writing—original draft, M.M.; Writing—review & editing, M.S. and M.M.; All authors have read and agreed to the published version of the manuscript.

**Funding:** This research received no external funding.

**Institutional Review Board Statement:** Not applicable.

**Informed Consent Statement:** Not applicable.

**Data Availability Statement:** The data that support the findings of this study are available from the corresponding author, upon reasonable request.

**Acknowledgments:** The authors would like to thank Editor-in-Chief, Editor and anonymous reviewers for their valuable reviews.

**Conflicts of Interest:** The authors declare no conflict of interest.

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
