# Peer review of "Numerical Study on Seismic Behavior of Flexural Frames with Semi-Rigid Welded Steel Connections Considering Static and Reciprocating Loads: A Performance-Based Earthquake Approach"

_applsci, doi:10.3390/app12157617_

Round 1

Reviewer 1 Report

1. Some of the figures are not legible, and the resolution must be improved, for instance, Figure 6. 2. Some models and modelling methods are employed in Section 2.1, while how they are used in the analysis is not clear, please provide a flow chart for the nonlinear static and dynamic analysis, and the describe the principles of the solvers.

3. An in-depth analysis of the cause of the failure of the removed beam should be made according to the load acting on the structure and the stresses distribution.

Author Response

Response to the reviewers’ comments (Reviewer #1)

The authors would like to thank the editor and the reviewers for their constructive criticisms and insightful comments on our paper. We have now substantially revised the paper addressing the reviewers’ comments, and hope that the revised paper is now suitable of publication in the journal of Applied Sciences. The corrections are highlighted with yellow color in the manuscript. 

  1. Some of the figures are not legible, and the resolution must be improved, for instance, Figure 6.2

Response: Thank you for pointing this out. The resolution of the figures as well as Fig.6.2 has now been improved.

2- Some models and modelling methods are employed in Section 2.1, while how they are used in the analysis is not clear, please provide a flow chart for the nonlinear static and dynamic analysis, and the describe the principles of the solvers.

Response: Thanks for your valuable comment. To explain this raised issue, we have expanded the section 2 to describe the methodology of the work in details.  As explained, the main model used in this study was the Ibarra-Krawwinkler model which was introduced to Open Sees software in order to establish a three-line curve named as constitutive curve. This curve is then used to calculate the nonlinear stiffness of the rotational springs by which the connections used in this study were modeled in the Open Sees software. 

  1. An in-depth analysis of the cause of the failure of the removed beam should be made according to the load acting on the structure and the stresses distribution.

Response: Referring to the subsection 3.2.1 (Types of connections used for modeling), for the second type of connection used in this study as shown in Fig.6, the formation of plastic hinges away from the beam end has been discussed in detail. Connection was modeled in Open Sees software, and its seismic behavior was investigated as discussed in the subsection 3.2.6 (Results and discussions). This study has focused only on the behavior of the connection model. However, the investigation of stress distribution in the beam or column was the beyond scope of the present study, and it should be investigated in the future studies which we are planning to do.   

Reviewer 2 Report

All results are right, but there are not protypal and original elements in the paper. 

Applsci-1749014 paper has several  disadvantages, which have to be corrected by authors.

The first disadvantage is the fact where the main target of the article is not clear (in reality, it is dark). Also, the documentation of the main issue (that is under question)  of the paper is inadequate. 

Thus, authors must rewrite the introduction.

Second, the methodology that is going to apply in order to examine the main issue of the paper must be given with pure/clear way (using bullets if it possible). Afterwards, the methodology must be applying, step-by step. 

Also, the Introduction is too big (about 2.5 pages).

Author Response

Please regard the attached file

Reviewer 3 Report

The paper "Investigation of the behavior of semi-rigid welded steel connections under static and reciprocating loads and their effect on the seismic behavior of steel flexural frames" is interesting and may be published after necessary corrections:

(1) The title is confusing to readers, it should be reformulated.;

(2) The abstract needs more quantitative information on the results found;

(3) The authors can provide more evidence at the end of the introduction of the innovation and final quality of this paper compared to other studies already published;

(4) Figures and equations should be of better quality, especially with internal texts. Also remove contour box in each one;

(5) Some results presented need to be further discussed by the authors, note that it is necessary to compare results with others in the literature;

(6) The conclusion is confusing and should not be joined with the abstract, the authors should separate these sections and the conclusion should be objective and short!

Author Response

Please regard the attached file

Round 2

Reviewer 2 Report

No comments.

Reviewer 3 Report

Ok.